# Genome-Wide Identification of bZIP Transcription Factors in Faba Bean Based on Transcriptome Analysis and Investigation of Their Function in Drought Response

**DOI:** 10.3390/plants12173041

**Published:** 2023-08-24

**Authors:** Lin-Tao Huang, Chang-Yan Liu, Li Li, Xue-Song Han, Hong-Wei Chen, Chun-Hai Jiao, Ai-Hua Sha

**Affiliations:** 1College of Agriculture, Yangtze University, Jingzhou 434025, China; hlt1406859197@outlook.com; 2Hubei Collaborative Innovation Center for Grain Industry, Jingzhou 434025, China; 3Engineering Research Center of Ecology and Agricultural Use of Wetland, Ministry of Education, Jingzhou 434025, China; 4Institute of Food Crops, Hubei Academy of Agricultural Sciences, Wuhan 430063, China; liucy0602@163.com (C.-Y.L.); hslili07@163.com (L.L.); hxs.1204@163.com (X.-S.H.); hwchen25@126.com (H.-W.C.); 5Key Laboratory of Crop Molecular Breeding, Ministry of Agriculture and Rural Affairs, Wuhan 430063, China; 6Hubei Key Laboratory of Food Crop Germplasm and Genetic Improvement, Wuhan 430063, China

**Keywords:** faba bean (*Vicia faba* L.), bZIP transcription factor, drought, ectopic expression, protein-protein interaction

## Abstract

Faba bean is an important cool-season edible legume crop that is constantly threatened by abiotic stresses such as drought. The basic leucine zipper (bZIP) gene family is one of the most abundant and diverse families of transcription factors in plants. It regulates plant growth and development and plays an important role in the response to biotic and abiotic stresses. In this study, we identified 18 members of the faba bean bZIP transcription factor family at the genome-wide level based on previous faba bean drought stress transcriptome sequencing data. A phylogenetic tree was constructed to group the 18 VfbZIP proteins into eight clades. Analysis of *cis*-acting elements in the promoter region suggested that these 18 *VfbZIPs* may be involved in regulating abiotic stress responses such as drought. Transcriptome data showed high expression of seven genes (*VfbZIP1*, *VfbZIP2*, *VfbZIP5*, *VfbZIP7*, *VfbZIP15*, *VfbZIP17*, and *VfbZIP18*) in the drought-tolerant cultivar under drought stress, in which *VfbZIP1*, *VfbZIP2*, and *VfbZIP5* were consistently expressed as detected by quantitative real-time polymerase chain reaction (qRT-PCR) compared to the transcriptome data. Ectopic overexpression of the three *VfbZIPs* in tobacco, based on the potato Virus X (PVX) vector, revealed that *VfbZIP5* enhanced the drought tolerance. Overexpressed *VfbZIP5* in plants showed lower levels of proline (PRO), malondialdehyde (MDA), and peroxidase (POD) compared to those overexpressing an empty vector under 10 days of drought stress. Protein-protein interaction (PPI) analysis showed that VfbZIP5 interacted with seven proteins in faba bean, including VfbZIP7 and VfbZIP10. The results depict the importance of *VfbZIPs* in response to drought stress, and they would be useful for the improvement of drought tolerance.

## 1. Introduction

Plants are difficult to escape from biotic and abiotic stresses such as pathogens, drought, cold, and heat due to their sedentary nature [1]. Abiotic stresses are major limiting factors in plant growth and cause significant economic losses around the world [2,3]. Among the various abiotic stresses, drought is one of the most serious threats to crop growth and production [4]. Plants evolved a range of mechanisms to adapt to or avoid adverse environmental factors involving stress perception, signaling, and the expression of specific stress-related genes and metabolites [5,6,7,8].

Transcription factors (TFs) are important components of the plant signaling pathway and determine the plant’s response to abiotic stresses [9]. The families of transcription factors can be defined based on sequences of the characteristic DNA-binding domains (DBDs) [10]. Plant transcription factors have many families such as DREB [11], WRKY [12], MYB [13], NAC [14], and bZIP [15]. These families of transcription factors are involved in regulating tolerance mechanism to various abiotic stresses in plants. For instance, the overexpression of *HsDREB1A* in bahiagrass enhances its tolerance to severe salt and drought stress [16]. Overexpression of *EaDREB2* enhances tolerance to drought and salinity in sugarcane [17]. The WRKY gene *SlDRW1* is a positive regulator of the defense response to grey mould and oxidative stress in tomatoes [18]. Overexpression of *MdSIMYB1* increases the tolerance of transgenic tobacco and apple to a variety of abiotic stresses [19]. Overexpression of wheat (*Triticum aestivum*) *TaNAC2a* improves plant drought tolerance [20]. A total of 320,370 transcription factor families from 165 species have been identified in the plant kingdom and are included in the PlantTFDB database, which is constantly updated [21,22].

The basic leucine zipper (bZIP) gene family is one of the most widespread and diverse families of transcription factors in plants [23]. The bZIP transcription factor family is named for its highly conserved bZIP structural domain, which consists of two structural features located on a contiguous alpha-helix. One is the basic amino acid region that binds to a specific DNA sequence, and the other is the leucine zipper, an amphipathic alpha-helix structure formed by a heptapeptide repeat sequence of leucine or other hydrophobic amino acids [15,24]. Many bZIP transcription factor families have been identified in different plant species such as *Arabidopsis* (78 *AtbZIPs*), soybean (160 *GmbZIPs*), rice (86 *OsbZIPs*), maize (125 *ZmbZIPs*), and wheat (227 *TabZIPs*) [25,26,27,28,29]. Functional characterization of bZIP transcription factors has been conducted in multiple plant species. For instance, *ZmbZIP22* is a regulator of maize endosperm development by regulating genes encoding storage protein 27-kDγ-zein [30]. In wheat, TubZIP28 and TabZIP28 proteins promote starch synthesis [31]. The bZIP transcription factor VdAtf1 regulates virulence by mediating nitrogen metabolism in *Verticillium dahlia* [32,33]. *Arabidopsis* IDD14 forms a functional complex with the bZIP transcription factors ABFs/AREBs and positively regulates abscisic acid (ABA)-mediated drought tolerance [34]. In the realm of legumes, the function of the bZIP gene has been predominantly studied in soybeans in response to drought and other stress such as *GmbZIP1, GmbZIP2*, *GmbZIP15, GmbZIP19, GmbZIP44*, *GmbZIP62*, *GmbZIP78, GmbZIP67*/*GsbZIP67*, *GmbZIP110*, *GmbZIP132, GmbZIP152*, and *GmFDL19* [35,36,37,38,39,40,41,42,43,44,45]. Moreover, *MtFbZIP1* has been found to respond to zinc deficiency response [46]. These findings demonstrate the participation of the bZIP gene family in the regulation of plant growth and development and its important role in response to biotic and abiotic stresses.

Faba bean (*Vicia faba* L.) is the third most important cool-season edible legume crop in the world and is grown in over 60 countries and territories [47]. Faba bean is a drought-sensitive crop, and the growth and yield can be severely affected by drought stress [48,49]. Faba bean requires abundant water to germinate. Non-uniform germination or non-germinating seed is usually caused by severe drought after sowing. Accordingly, serious production reductions or no harvest occurred. Although there were some reports on drought tolerance in faba bean [50,51], effective means to improve drought tolerance are lacking and little is known about the mechanisms of drought tolerance in faba bean. In this study, 18 bZIP transcription factor members were identified in faba bean at the genome-wide level based on the sequencing of a faba bean drought stress transcriptome. The physicochemical properties, gene structure, protein structural domains, conserved motifs, *cis*-acting elements, and evolutionary relationships were analyzed through the bioinformatical method. Three *VfbZIPs* were selected for biological function analysis for their role in drought stress response based on the results of RNA-Seq and quantitative real-time polymerase chain reaction (qRT-PCR), and *VfbZIP5* was validated to be able to increase drought tolerance of tobacco by ectopic overexpression. These findings provide insights into the function of *VfbZIPs* in response to drought stress, and they could be used in the breeding program for the improvement of drought tolerance.

## 2. Results

### 2.1. Identification of VfbZIP Genes in Faba Bean

Eighteen bZIP genes were identified in faba bean and they were designated as VfbZIP1-VfbZIP18. Protein multiple sequence alignment (Figure 1A) and conserved domain analysis (Figure 1C) showed that all 18 VfbZIP proteins contained a basic region leucine zipper (BRLZ) domain. With the exception of VfbZIP1, VfbZIP11, VfbZIP12, VfbZIP14, and VfbZIP18, the majority of the VfbZIP proteins have the low complexity region domain (LCD). VfbZIP1 and VfbZIP12 possess a transmembrane domain (TMD) and only VfbZIP5 contains a coiled coil region domain (CCD).

### 2.2. Physical and Chemical Characteristics of VfbZIPs

The proteins encoded by the *VfbZIP* genes ranged from 114 to 557 amino acids in length, with molecular weights of 13.15 to 61.71 kDa, isoelectric points of 5.37 to 10.57, and aliphatic indexes of 51.16 to 86.99 (Table 1). All instability indexes were greater than 40, indicating that these are unstable proteins. Similarly, all grand averages of hydropathicity (GRAVY) were negative, indicating that these are hydrophilic proteins. The subcellular localization of all bZIP proteins was in the nucleus, which is consistent with the characteristics of transcription factors. The secondary structures of all bZIP proteins were further provided in Appendix A.

### 2.3. Phylogenetic Analysis of VfbZIPs

To better understand the evolutionary relationships among bZIP proteins from different species, we constructed a rootless phylogenetic tree using 235 bZIP proteins from six species. The accession of all proteins is presented in Appendix A. The phylogenetic tree was divided into eight subfamilies (Figure 2). The Subfamily H contained the highest number of VfbZIPs, with five VfbZIPs and 64 other bZIPs. The second and third largest subfamilies were B and D, containing a total of 41 and 40 bZIPs, respectively, which included two and three VfbZIPs. Subfamilies A, C, E, F, and G contained two, one, two, one, and one VfbZIPs, respectively.

### 2.4. Gene Structure and Motif Distribution

The gene structures of *VfbZIPs* were analyzed (Figure 3B). Among the 18 *VfbZIPs*, *VfbZIP14* and *VfbZIP18* contained the highest number of exons, with a total of 12 exons each, while *VfbZIP13* and *VfbZIP17* had only one exon. The remaining *VfbZIPs* had between two and seven exons: *VfbZIP1* had two exons, *VfbZIP9*, *VfbZIP12*, and *VfbZIP15* had three exons, *VfbZIP2*, *VfbZIP3*, *VfbZIP6*, *VfbZIP8*, *VfbZIP10*, and *VfbZIP11* had four exons, *VfbZIP5* and *VfbZIP7* had five exons, *VfbZIP16* had six exons, and *VfbZIP4* had seven exons.

Ten motifs were found within the 18 *VfbZIPs* according to the motif analysis (Figure 3C). All members contained motif 1 (BRLZ domain), indicating that the BRLZ domain is highly conserved in the *VfbZIP* genes. Additionally, the members with the closer evolution showed more similar conserved motifs. For instance, *VfbZIP10* and *VfbZIP15* shared motif 1 and motif 7, while *VfbZIP7* and *VfbZIP8* shared motif 1, motif 2, motif 3, motif 4, motif 5, motif 6, motif 8, and motif 10. These motifs also exhibited similar distribution and length. The sequences of the conserved motifs are displayed in Appendix A.

### 2.5. Cis-Acting Elements of the VfbZIP Genes

As the 18 *VfbZIP* genes were identified based on the transcriptome and the untranslated regions (UTRs) information was lacking in the genome, we first mapped them to the public reference genome by aligning the protein sequences (Appendix A). Then the 2000 bp sequences upstream of the start codon in the 18 *VfbZIP* genes were analyzed to identify *cis*-acting elements. The *VfbZIP* genes contained a variety of stress response elements, and the largest number of 15 abiotic stress-related elements were selected for further analysis (Figure 4B and Appendix A). These elements include the abscisic acid response element (ABRE), eight light response elements (AE-box, Box 4, GATA-motif, G-Box, GT1-motif, I-box, TCT-motif, and MRE), the anaerobic induction element (ARE), two methyl jasmonate (MeJA) response elements (CGTCA-motif and TGACG-motif), the drought response element (MBS), the salicylic acid (SA) response element (TCA-element), and the auxin response element (TGA-element). The analysis revealed that each gene featured a minimum of seven *cis*-acting elements linked to abiotic stress. Notably, all *VfbZIP* genes contained G-Box. Among the abiotic stress-related elements identified, ABRE, ARE, Box 4, CGTCA-motif, and TGACG-motif were the most recurrent ones. The findings suggest that the 18 *VfbZIP* genes may be involved in the response to abiotic stress.

### 2.6. Differential Expression of VfbZIPs under Drought Stress

To identify whether the *VfbZIP* genes involved in drought stress response, we analyzed their expression levels in drought-tolerant (CDAS105) and drought-sensitive (E−Can No. 1) varieties after 16 and 64 h of germination under drought treatment using the transcriptome data (Figure 5). Five genes (*VfbZIP2*, *VfbZIP7*, *VfbZIP15*, *VfbZIP17*, and *VfbZIP18*) were highly expressed in CDAS105 after 16 h of germination under drought stress (T2−16), while two genes (*VfbZIP1* and *VfbZIP5*) were highly expressed after 64 h of germination under drought stress (T2−64). Those genes may positively regulate drought response in faba bean. Conversely, one gene (*VfbZIP10*) highly expressed in E−Can No. 1 after 64 h of germination under drought stress (T1−64), which may negatively regulate drought response in faba bean. The expression levels of these eight *VfbZIP* genes were further validated by qRT-PCR (Figure 6). The expression patterns of *VfbZIP1* and *VfbZIP5* were consistent with the transcriptome data, as they were significantly expressed in T2−64. Additionally, *VfbZIP2* displayed high expression levels in both T2−16 and T2−64, although it was expressed highest in T2−16 as detected in transcriptome data. These three genes might be associated with the drought tolerance of the variety CDAS105, so they were selected for further biological function investigation.

### 2.7. Validation of the Biological Function of VfbZIPs

The potato Virus X (PVX) vector has the ability to highly express foreign genes, and the expressed product can move from initially inoculated cells to uninfected sites. As a result, the PVX vector has been successfully used in investigating the functions of genes involved in drought stress [52]. In this study, we validated the functions of *VfbZIP1*, *VfbZIP2*, and *VfbZIP5* by overexpressing them in N. *benthamiana* using the PVX vector. Only the ectopic overexpression of *VfbZIP5* was found to increase the drought tolerance of tobacco plants. Before the drought treatment, both the empty vector overexpressed (EV) and *VfbZIP* overexpressed (*VfbZIP*-OE) tobacco seedlings exhibited similar growth phenotypes. After 15 days of drought treatment, *VfbZIP5*-OE plants exhibited better growth than EV plants, with significantly less leaf wilting, curling, water loss, and fading green, indicating a drought-tolerant phenotype (Figure 7A). The reverse transcription-polymerase chain reaction (RT-PCR) results showed that *VfbZIP5* was expressed in *VfbZIP5*-OE plants, while it was not expressed in EV plants (Figure 7B).

The proline (PRO), malondialdehyde (MDA), peroxidase (POD), and superoxide dismutase (SOD) levels are important indicators of the effects of abiotic stress on plant growth. PRO is a protective agent against osmotic stress in plants, and the level of MDA can indicate the extent of membrane lipid peroxidation damage. Plants create a huge number of reactive oxygen species (ROS) under unfavorable conditions, which antioxidant enzymes like POD and SOD can scavenge the ROS. We investigated the levels of PRO, MDA, POD, and SOD in EV and *VfbZIP*-OE tobacco strains before and after drought stress in order to better understand the physiological mechanisms of overexpressing the *VfbZIP5* gene to promote drought tolerance. According to the results (Figure 7C–F), there were no physiological changes between the EV and *VfbZIP*-OE strains prior to drought stress treatment. The PRO content, MDA content, and POD activity of the *VfbZIP5*-OE plant were significantly lower than those of the EV strain under drought stress, while the SOD activity was not statistically significantly different from that of the EV strain. These findings imply that overexpressing *VfbZIP5* might increase the plant’s drought tolerance by decreasing the membrane lipid peroxidation damage.

### 2.8. Protein-Protein Interaction Network Prediction

To comprehensively understand the potential interactions and functions of VfbZIP5 with other proteins in faba bean, we used the STRING database to map the interactions of VfbZIP5 with other proteins in faba bean using soybean (*Glycine max*) as a reference species. Five faba bean proteins (TRINITY_DN23524_c0_g1, TRINITY_DN5292_c0_g1, TRINITY_DN18484_c0_g1, TRINITY_DN8620_c0_g1, and TRINITY_DN13870_c0_g1) directly interacted with VfbZIP5 (Figure 8 and Appendix A). Intriguingly, two other VfbZIPs (VfbZIP7 and VfbZIP10) were detected in the protein-protein interaction (PPI) network, implying there were associated functions among different VfbZIPs. The predicted interaction proteins can help to further understand the mechanism by which VfbZIP5 is involved in the regulation of drought response in faba bean.

## 3. Discussion

### 3.1. VfbZIP Gene Family Identification and Analysis of Cis-Acting Elements

The bZIP family genes and their role have not been fully investigated in faba bean. In this study, we identified 18 faba bean bZIP genes based on prior faba bean drought stress transcriptome sequencing data. All *VfbZIPs* are subcellularly localized in the nucleus, consistent with transcription factor characteristics (Table 1). All 18 genes were successfully mapped to the public reference genome (Appendix A) [53]. The *VfbZIP* genes contain a number of *cis*-acting elements associated with abiotic stress (Figure 4B and Appendix A), including the ABRE and MBS elements, which play an important role in the response to drought stress [54,55]. Additionally, there are four hormone response elements such as the CGTCA-motif, eight light response elements including Box 4, G-Box, and I-Box, and anaerobic induction element ARE. These findings imply that *VfbZIP* genes may regulate abiotic stress reactions such as drought.

### 3.2. Phylogenetic Analysis of VfbZIPs

The 18 VfbZIP proteins were divided into 8 subfamilies (Figure 2), in which VfbZIP13 and VfbZIP17 were categorized into subfamily A with AtbZIP11 and AtbZIP53, while VfbZIP16 was categorized into subfamily C with AtbZIP9. These proteins are part of the C/S1 bZIP network found in *Arabidopsis*, which is a signaling hub that promotes metabolic redistribution and is linked to adversity tolerance [25]. VfbZIP14 and VfbZIP18, together with AtbZIP41/GBF1, AtbZIP54/GBF2, and AtbZIP55/GBF3, were classified as subfamily B. G-box binding factor (GBF)-type bZIP transcription factors have been reported to be involved in plant responses to hormones like abscisic acid, which are crucial for plant tolerance [56]. Furthermore, it has been noted that GBF1 controls *CAT2* and *PAD4* to promote disease tolerance in *Arabidopsis* and function as a factor in the process upstream of salicylic acid accumulation [57]. VfbZIP9, VfbZIP10, and VfbZIP15 were categorized as subfamily D and are homologous to AtbZIP36, AtbZIP66, and AtbZIP40/GBF4, respectively. AtbZIP36 and AtbZIP66 are abscisic acid responsive element-binding factor (ABF/AREB)-type bZIP transcription factors that are linked to plant development and major stress responses, such as drought and salt stress [58]. AtbZIP40/GBF4 is a GBF-type bZIP transcription factor that, as previously mentioned, may be involved in plant stress tolerance responses via hormonal routes such as abscisic acid. VfbZIP1, VfbZIP12, and AtbZIP28 were grouped into subfamily E. AtbZIP28 is a converter of the unfolded protein response (UPR), which may be connected to high temperature and stress response [59]. VfbZIP4 was placed in subfamily F, and its homologue AtbZIP56 is an ELONGATED HYPOCOTYL5 (HY5) type bZIP transcription factor. HY5 has been shown to limit hypocotyl growth and lateral root development while promoting pigment accumulation. Furthermore, HY5 serves as a regulatory hub for signaling pathways involving hormones, nutrition, abiotic stress, and reactive oxygen species [60]. VfbZIP6 was categorized as subfamily G, and research suggests that AtbZIP34, a homolog of VfbZIP6, is essential for controlling pollen wall construction and pollen growth in *Arabidopsis thaliana* [61]. Subfamily H contains the most VfbZIP members, with six (VfbZIP2, VfbZIP3, VfbZIP5, VfbZIP7, VfbZIP8, and VfbZIP11). In *Arabidopsis*, subfamily I bZIP proteins play regulatory functions in stress response, cell cycle regulation, and other aspects of development [25], with AtbZIP51/VIP1 (a homolog of VfbZIP11) identified as an osmosensing signal regulator [62]. These findings suggest a correlation between stress tolerance and 18 members of the faba bean bZIP family.

### 3.3. Analysis of the VfbZIP5 Drought Tolerance Function

Seven genes (*VfbZIP1*, *VfbZIP2*, *VfbZIP5*, *VfbZIP7*, *VfbZIP15*, *VfbZIP17*, and *VfbZIP18*) displayed high expression in the drought-tolerant cultivar under drought stress in transcriptome data, and the expression of *VfbZIP1*, *VfbZIP2*, *VfbZIP5* detected by qRT-PCR were consistent with the transcriptome data (Figure 5 and Figure 6). Further functional investigation of *VfbZIP1*, *VfbZIP2*, and *VfbZIP5* by ectopically overexpressing them in tobacco revealed that only *VfbZIP5* can enhance the drought tolerance of tobacco (Figure 7A). The content of PRO/MDA and the activity of POD in the *VfbZIP5*-OE were significantly decreased compared to the control under drought stress. Higher levels of PRO can enhance plant tolerance to drought [63]. MDA is commonly used to indicate membrane lipid peroxidation injury in stressed plants [64]. POD is an important antioxidant that neutralizes ROS, which are highly reactive radicals produced in response to abiotic stresses in plants [37,65]. The decreased levels of MDA and POD may regulate membrane lipid peroxidation and ROS neutralization to enhance the drought tolerance of *VfbZIP5*-OE, whereas the PRO level is not tightly related to drought tolerance. By comparing *VfbZIP5* with bZIP genes with function identification in other legumes based on construction of phylogenetic tree [66], *VfbZIP5* can cluster with *GmbZIP19* and *MtFbZIP1* (Appendix A and Appendix A). *GmbZIP19* was reported to negatively impacts drought tolerance [38], whereas *VfbZIP5* positively regulate drought tolerance. It implies that genes with close phylogenetic relationships may share similar or reverse functions.

### 3.4. Protein-Protein Interaction Network Prediction

The PPI analysis revealed that five faba bean proteins directly interacted with VfbZIP5 (Figure 8 and Appendix A). Among them, the homologous proteins of TRINITY_DN5292_c0_g1 and TRINITY_DN8620_c0_g1 in soybean were annotated as F-box Tubby-like proteins (TLP). TLP is known to play a key role in plant growth and development and in response to biotic and abiotic stresses. For instance, overexpression of *GmTLP8* enhances the tolerance of soybean to drought and salt stress [67]. In *Arabidopsis*, AtTLP2 plays a complex role in ABA-dependent abiotic stress signaling, especially in response to salt and dehydration stresses [68]. The homologous protein of TRINITY_DN18484_c0_g1 in soybean was annotated as the basic helix-loop-helix (bHLH) transcription factor protein. The bHLH transcription factor family is known to be one of the largest transcription factor gene families in *Arabidopsis*, with pleiotropic regulatory effects on plant growth and development, stress response, metabolic function, and signal network [69]. Additionally, VfbZIP7 and VfbZIP10 were also detected in the PPI network. The homologous protein of VfbZIP7 in soybean was annotated as an RF2b-type bZIP transcription factor. RF2b has been reported to be involved in the occurrence and development of rice tungro disease symptoms [70]. The homologous protein of VfbZIP10 in soybean was annotated as an AREB-type bZIP transcription factor. As mentioned above, the AREB-type bZIP transcription factor plays a central role in plant development and major stress responses such as drought and salt stress. These results indicate that VfbZIP5 may regulate drought stress by interacting with other stress-responsive proteins.

## 4. Materials and Methods

### 4.1. Identification of the bZIP Gene Family in Faba Bean

In previous work, we obtained a reference transcript genome of faba bean by combining Pacbio SMRT sequencing and Illumina sequencing, and the data were loaded in DDBJ/ENA/GenBank (accession number: GISP01000000). Based on the drought stress transcriptome sequencing data, the genes annotated as bZIP were screened using the NR, Swiss-prot, and COG databases, and the corresponding protein sequences were extracted using TBtools software (v1.108) [71]. To remove the non-bZIP gene sequences, repetitive sequences, and redundant transcript data, we utilized online tools such as SMART “https://smart.embl.de/ (accessed on 7 July 2022)”, NCBI CDD “https://www.ncbi.nlm.nih.gov/cdd/ (accessed on 7 July 2022)”, and PFAM “http://pfam.xfam.org/ (accessed on 7 July 2022)” to predict sequence conserved structural domains. The identified genes were named as *VfbZIP1* to *VfbZIP18* based on the order of transcriptome sequencing nomenclature.

### 4.2. Physicochemical Properties, Protein Secondary Structure, and In-Silico Subcellular Localization of VfbZIPs

The 18 VfbZIP protein sequences identified were submitted to the online website ExPASy “https://web.expasy.org/protparam/ (accessed on 25 November 2022)” to analyze various physicochemical properties such as the molecular weight, isoelectric point, protein hydrophilicity, and instability coefficient. The secondary structure of the proteins was predicted using the online website SOPMA “https://npsa-prabi.ibcp.fr/cgi-bin/npsa_automat.pl?page=npsa_sopma.html (accessed on 25 November 2022)”, while the *in-silico* subcellular localization analysis was carried out on the Plant-mPLoc website “http://www.csbio.sjtu.edu.cn/bioinf/plant-multi/# (accessed on 25 November 2022)”.

### 4.3. Multiple Sequence Alignment and Phylogenetic Analysis

As reported in previous studies [25,26,27,28,29], the bZIP protein sequences of *Arabidopsis* (AtbZIP), soybean (GmbZIP), rice (OsbZIP), maize (ZmbZIP), and wheat (TabZIP) were downloaded from the online website Ensembl Plants “http://plants.ensembl.org/ (accessed on 25 November 2022)”. The default parameters of the ClustalW program in MEGA software (v7.0.21) [72] were used for the sequence alignment of bZIP proteins from faba been, *Arabidopsis thaliana*, soybean, rice, maize, and wheat. A phylogenetic tree was constructed using the neighbor-joining (NJ) method with 1000 bootstrap method, poisson model, and pairwise deletion. The iTOL website “https://itol.embl.de/ (accessed on 27 November 2022)” was then used to enhance the visual appeal of the phylogenetic tree.

The ClustalW program was also utilized to conduct multiple sequence comparisons of the 18 VfbZIP protein sequences as mentioned above. The resulting files were then submitted to DNAMAN 7.0 software (Lynnon Biosoft, San Ramon, CA, USA) for the purpose of visualization.

### 4.4. Gene Structure, Motif, and Protein Conserved Domain of VfbZIPs

We utilized the online tool MEME “https://meme-suite.org/ (accessed on 11 March 2023)” to analyze the motif types and orders of bZIP family proteins in faba been, with a set number of motif predictions at ten. Additionally, we employed SMART to analyze the conserved domains of the proteins and visualized the results using TBtools software (v1.108). We blasted the protein sequences of the 18 *VfbZIP* genes using TBtools software (v1.108) with reference to the recently published faba bean whole genome data [53]. We then utilized TBtools software (v1.108) to analyze and visualize the structure of the *VfbZIP* genes.

### 4.5. Analysis of Cis-Acting Elements in VfbZIP Gene Promoters

We extracted 2000 bp promoter sequences upstream of the 18 *VfbZIP* genes from the whole genome data of faba bean published by Jayakodi et al., 2023 [53] using TBtools software (v1.108). These promoter sequences were subjected to *cis*-acting elements analysis using the online program PlantCARE “http://bioinformatics.psb.ugent.be/webtools/plantcare/html/ (accessed on 12 March 2023)”, and the results were visualized using TBtools (v1.108).

### 4.6. Transcriptome Expression Pattern of VfbZIPs

Using the faba bean drought stress transcriptome sequencing data (DDBJ/ENA/GenBank, accession number: GISP01000000), we analyzed the expression of 18 *VfbZIPs* in two faba bean varieties that were subjected to drought treatment for 16 and 64 h. The results were visualized using TBtools software (v1.108).

### 4.7. Plant Materials and Treatments

The drought-sensitive faba bean variety E−Can No. 1 was bred by the Hubei Academy of Agricultural Sciences, and the drought-tolerant variety CDAS105 was introduced from Ethiopia. The uniform size seeds of both varieties were disinfected in 10% hypochlorite for five minutes and then rinsed four times with sterile water. Next, twenty-five treated seeds were placed in 12 cm diameter Petri dishes containing two layers of filter paper and soaked in 15 mL of 10% mannitol solution for three repetitions. Distilled water-treated seeds were used as the control. The Petri dishes were incubated in a growth chamber with a constant temperature of 25 °C for 16 h of light and 8 h of darkness. At 16 and 64 h after germination, 1/4 of the embryonic cotyledons were removed from the all seeds, frozen in liquid nitrogen, and stored at −80 °C for qRT-PCR analysis. Three biological replicates were set.

### 4.8. RNA Extraction and qRT-PCR Analysis

Total RNA was extracted from the plant using the TRNzol reagent method (TIANGEN, Beijing, China). The complementary DNA (cDNA) was synthesized using a reverse transcription kit (Promega Corporation, Beijing, China) according to the manufacturer’s protocol. Fluorescence quantification primers were designed using Primer Premier 5.0 (Appendix A). QRT-PCR was performed using the Color Fluorescent Quantitative Pre-mixture Kit (TIANGEN, Beijing, China) according to the instructions, with *VfNADHD4* of faba bean as the reference gene [73]. The experiment was conducted using the CFX96™ Real-Time PCR Detection System (Bio-Rad, Hercules, CA, USA). The reactions were conducted as follow: 95 °C for 15 min, followed by 40 cycles of 95 °C for 10 s and 57 °C for 20 s, and the melting curve stage. Each sample was tested in three technical and three biological replicates. The 2(-Delta Delta C(T)) method [74] was employed to determine the differences in the expression of the *VfbZIP* family genes. Data were analyzed using Microsoft Excel 2016 and SPSS 26.0, and graphs were created with GraphPad Prism (v8.0.2). Significance analysis was carried out using a one-way analysis of variance (ANOVA).

### 4.9. Ectopic Overexpression, RT-PCR, and Phenotypic Analysis

Four-week-old tobacco (*Nicotiana benthamiana* L.) plants were used for ectopic overexpression of the *VfbZIPs* as described by Han et al., 2021 [52]. The tobacco plants were grown in square-shaped plastic pots with 10 cm in diameter and 8.5 cm in height, with a 1:1 mix of vermiculite and nutrient soil. These pots were placed in trays with 100 cm in length and 30 cm in width. Each tray contained five pots of plant with the infiltration of targeted genes and the infiltration of empty vectors. Three biological replicates were set for each targeted gene. The trays were spaced 10 cm apart to ensure adequate light exposure and air circulation. All trays were housed within a greenhouse with a constant temperature of 24 °C, a 16:8 h light-dark ratio, 100 μM m^−2^ s^−1^ white light, and 70% relative humidity. Regular watering was conducted to maintain soil moisture levels.

The coding sequences of *VfbZIPs* were amplified using gene-specific primers (Appendix A), inserted into the PVX-LIC vector, and verified by sequencing. The recombinant vectors were introduced into *Agrobacterium tumefaciens* GV3101 using the freeze-thaw method, while the PVX-LIC empty vector served as the control, and all of them were injected into tobacco by the infiltration method. After seven days, RNA was extracted from newly expanded leaves that were not inoculated, using tobacco *actin* as the internal reference for RT-PCR analysis, as described by Sha et al., 2015 [75]. Concurrently, the plants were treated with a water withheld treatment, and their phenotype was identified and photographed 15 days after the treatment.

### 4.10. Physiological Parameter Measurements

The plants were subjected to water withheld when they were injected for 7 days, and the newly expanded leaves were collected at 0 days and 10 days after water withheld for physiological parameter measurement. The PRO and MDA contents, along with the activities of SOD and POD, were quantified using reagent kits according to manufacturer instructions (NanJing JianCheng Bioengineering Institute, China). Each experiment was performed with three technical and three biological replicates to ensure the reliability of the results. Data were analyzed using Microsoft Excel 2016 and SPSS 26.0, and graphs were created with GraphPad Prism (v8.0.2). Significance analysis was conducted using ANOVA.

### 4.11. Prediction of Protein-Protein Interaction Networks

PPI analysis provides a basis for predicting protein function. To investigate the potential interaction between VfbZIP proteins responsible for drought tolerance and other proteins in faba bean, we utilized soybean (*Glycine max* L.), which is also a legume, as a reference species and established a faba bean PPI network through the STRING database “https://string-db.org/ (accessed on 10 June 2023)”. We evaluated and predicted the protein interaction information of faba bean based on known protein interaction relationships. To enhance the visualization, we utilized Cytoscape software (v3.7.1) [76]. Additionally, we performed sequence alignment of soybean and faba bean proteins using TBtools (v1.108).

## 5. Conclusions

A total of 18 *VfbZIP* genes were identified at the whole genome level, and the ectopic overexpression of *VfbZIP5* improved drought tolerance in tobacco. Additionally, VfbZIP5 may interact with seven proteins in faba bean. The study provides a scientific basis for understanding the function of *VfbZIP5* in response to drought stress and the wider importance of *VfbZIPs* in plant abiotic stress responses. However, further research is required to precisely elucidate the involvement of *VfbZIPs* in abiotic stress responses.

## Figures and Tables

**Figure 1 plants-12-03041-f001:**
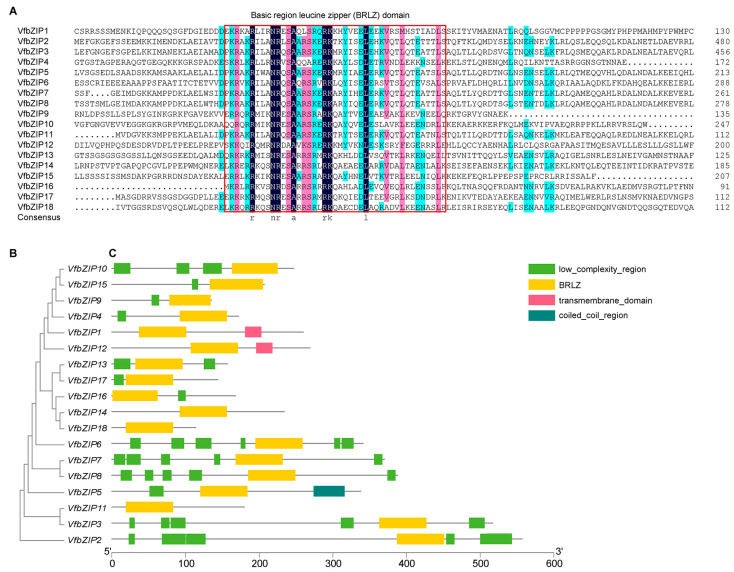
Protein multiple sequence alignment, rectangular phylogenetic tree, and conserved domain analysis of VfbZIPs. (**A**) Protein multiple sequence alignment result. Black, pink, and light blue shadings are used to represent 100%, 75%, and 50% amino acid similarities, respectively. The sequences in the red box are the basic region leucine zipper (BRLZ) conserved domains. (**B**) The phylogenetic tree. The ClustalW program was used to compare the full-length amino acid sequences of VfbZIP proteins. The phylogenetic tree was constructed by the neighbor-joining (NJ) method, and the bootstrap was repeated 1000 times. (**C**) Conserved domain analysis. Different color frames on the black line represent different domains.

**Figure 2 plants-12-03041-f002:**
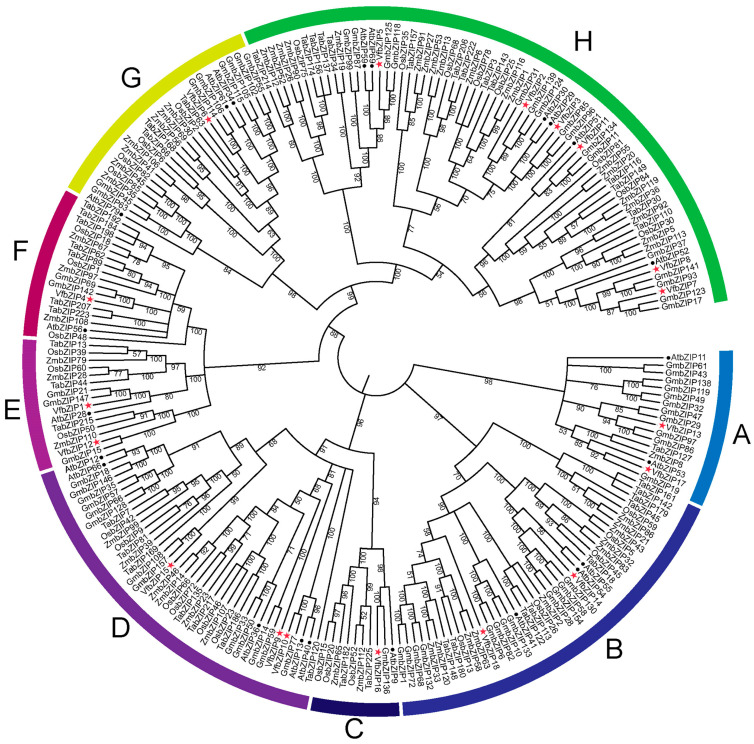
Phylogenetic analysis of the bZIP proteins. The ClustalW program was used to compare the full-length amino acid sequences of bZIP proteins in *Arabidopsis thaliana* (AtbZIP), soybean (GmbZIP), rice (OsbZIP), maize (ZmbZIP), wheat (TabZIP), and faba bean (VfbZIP). The phylogenetic tree was constructed by the neighbor-joining (NJ) method, and the bootstrap was repeated 1000 times. Different subfamilies are marked with different colors.

**Figure 3 plants-12-03041-f003:**
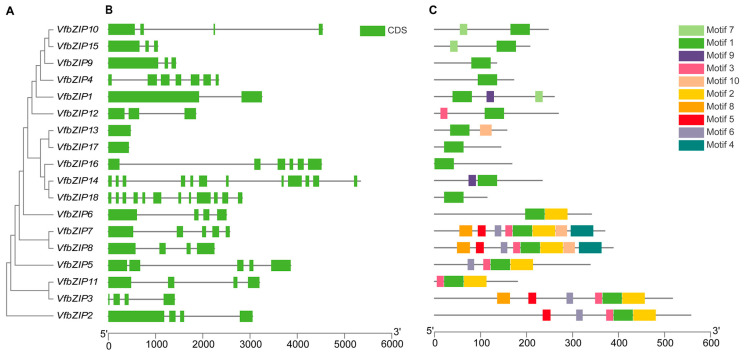
Rectangular phylogenetic tree, gene structure, and motif distribution of *VfbZIPs*. (**A**) The phylogenetic tree. (**B**) Gene structure. The green box and black line represent the exon and intron, respectively. (**C**) Motif distribution. The diagram shows 10 conserved motifs in the VfbZIP proteins, and different color boxes represent different motifs.

**Figure 4 plants-12-03041-f004:**
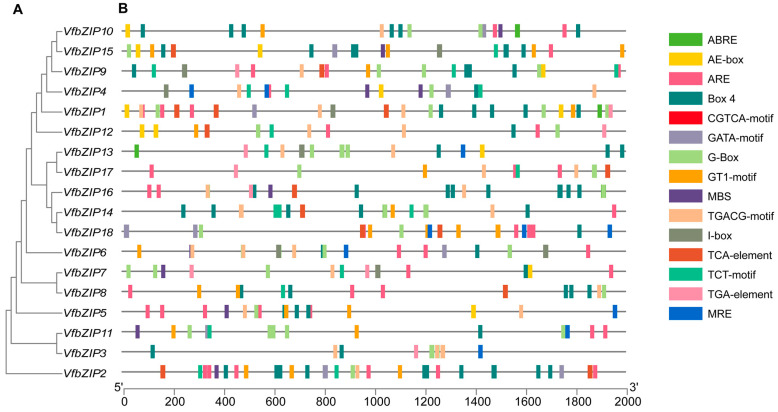
Rectangular phylogenetic tree and *cis*-acting elements in the promoter region of *VfbZIP* genes. (**A**) The phylogenetic tree. (**B**) Analysis of *cis*-acting elements in the promoter region of *VfbZIP* genes. Different color boxes on the black line represent different *cis*-acting elements.

**Figure 5 plants-12-03041-f005:**
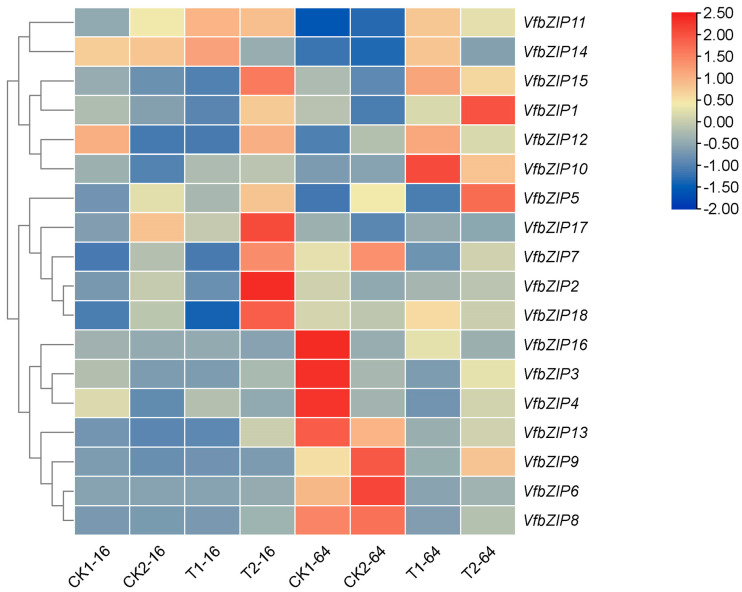
Expression analysis of *VfbZIP* genes based on transcriptome data. T1−16/T1−64 represents the drought-sensitive variety (E−Can No. 1) after 16 or 64 h of germination under drought stress. T2−16/T2−64 represents the drought-tolerant variety (CDAS105) after 16 or 64 h of germination under drought stress. CK1−16/CK1−64 represents the drought-sensitive variety (E−Can No. 1) after 16 or 64 h of germination under normal treatment. CK2−16/CK2−64 represents the drought-tolerant variety (CDAS105) after 16 or 64 h of germination under normal treatment. Red in the heat map indicates a higher level of expression. Blue indicates a lower level of expression.

**Figure 6 plants-12-03041-f006:**
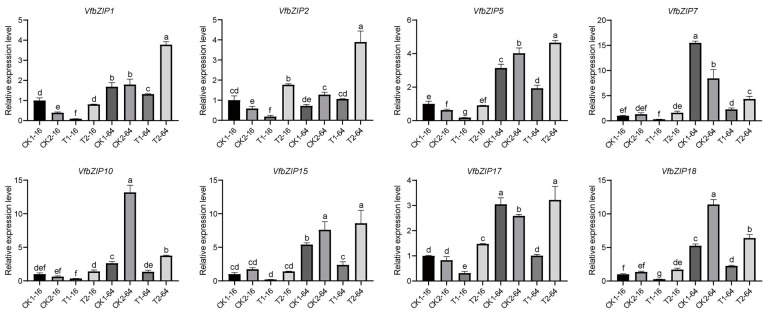
The expression levels of *VfbZIP* genes by quantitative real-time polymerase chain reaction (qRT-PCR) analysis. T1−16/T1−64 represents the drought-sensitive variety (E−Can No. 1) after 16 or 64 h of germination under drought stress. T2−16/T2−64 represents the drought-tolerant variety (CDAS105) after 16 or 64 h of germination under drought stress. CK1−16/CK1−64 and CK2−16/CK2−64 represents the two varieties after 16 or 64 h of germination under normal treatment. The y-axis represents the relative expression level, and the data were presented as the mean ± standard deviation (SD) of three technical and three biological replicates. One-way analysis of variance (ANOVA) was used for significance analysis, and the letters a to g represented significant differences at the *p* < 0.05 level.

**Figure 7 plants-12-03041-f007:**
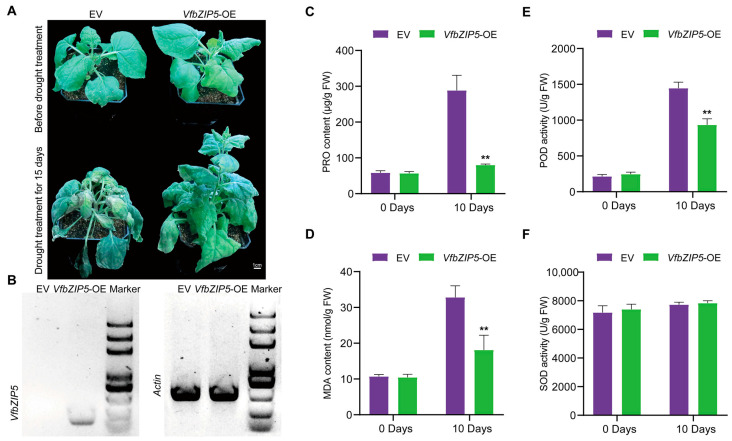
Biological function analysis of *VfbZIP5*. (**A**) Phenotypic identification of *VfbZIP5*-overexpressing tobacco plants (*VfbZIP5*-OE) and empty vector plants (EV) before and 15 days after drought stress. (**B**) The reverse transcription-polymerase chain reaction (RT-PCR) analysis of *VfbZIP5*-OE plants and EV plants. With the tobacco *actin* gene as a reference, the marker used was 5000 bp. Drought treatment 0 days and drought treatment 10 days, (**C**) proline (PRO) content, (**D**) malondialdehyde (MDA) content, (**E**) peroxidase (POD) activity, and (**F**) superoxide dismutase (SOD) activity in tobacco EV lines and *VfbZIP5*-OE lines. The data were presented as the mean ± standard deviation (SD) of three technical and three biological replicates. The significance analysis was performed by one-way analysis of variance (ANOVA). ** represents a very significant difference at the *p* < 0.01 level.

**Figure 8 plants-12-03041-f008:**
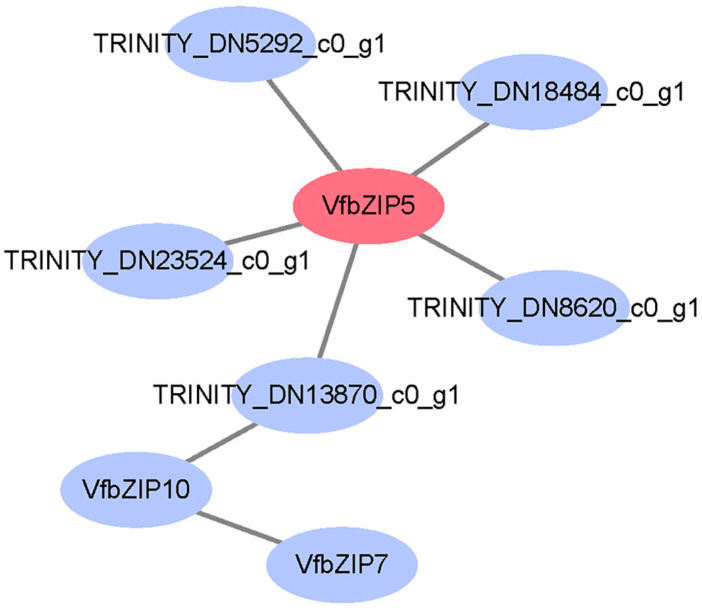
Protein-protein interaction (PPI) network prediction. The interaction network of VfbZIP5 protein in faba bean was constructed by using the STRING database with soybean (*Glycine max*) as reference species. The circular nodes in the network represent proteins, and the lines connecting the nodes represent interactions.

**Table 1 plants-12-03041-t001:** Basic information about bZIP in faba bean.

Protein Name	Transcriptome Sequencing ID	Length(aa)	MW (kDa)	PI	II	Aliphatic Index	GARVY	Subcellular Localization Prediction
VfbZIP1	TRINITY_DN1370_c0_g1	260	29.38	9.76	69.22	67.77	−0.643	Nuclear
VfbZIP2	TRINITY_DN2872_c0_g1	557	61.71	7.81	64.35	62.35	−0.878	Nuclear
VfbZIP3	TRINITY_DN3948_c0_g1	517	57.16	6.78	62.77	60.62	−0.934	Nuclear
VfbZIP4	TRINITY_DN4064_c0_g1	172	18.55	9.74	52.08	51.16	−1.251	Nuclear
VfbZIP5	TRINITY_DN7170_c0_g1	338	38.02	9.6	59.87	68.52	−0.868	Nuclear
VfbZIP6	TRINITY_DN8121_c0_g1	341	38.14	5.55	67.79	70.38	−0.779	Nuclear
VfbZIP7	TRINITY_DN8538_c0_g1	370	40.39	6.31	57.72	60.46	−0.786	Nuclear
VfbZIP8	TRINITY_DN9344_c0_g1	388	42.21	6.08	49.11	55.15	−0.868	Nuclear
VfbZIP9	TRINITY_DN9459_c0_g1	135	14.68	10.23	41.15	83.78	−0.527	Nuclear
VfbZIP10	TRINITY_DN16349_c0_g1	247	27.58	5.37	49.29	71.78	−0.661	Nuclear
VfbZIP11	TRINITY_DN19919_c0_g1	180	20.58	9.94	56.05	66.22	−0.968	Nuclear
VfbZIP12	TRINITY_DN22348_c0_g1	269	30.89	6.03	67.27	86.99	−0.412	Nuclear
VfbZIP13	TRINITY_DN24453_c0_g1	157	17.51	9.1	63.89	73.31	−0.621	Nuclear
VfbZIP14	TRINITY_DN26696_c0_g1	234	25.79	8.74	55.51	66.37	−0.916	Nuclear
VfbZIP15	TRINITY_DN26944_c0_g1	207	23.14	9.6	58.2	73.04	−0.785	Nuclear
VfbZIP16	TRINITY_DN30461_c0_g1	168	18.73	10.57	46.34	78.93	−0.652	Nuclear
VfbZIP17	TRINITY_DN31066_c0_g1	144	16.66	6.77	61.18	71.81	−1.031	Nuclear
VfbZIP18	TRINITY_DN41604_c0_g1	114	13.15	5.95	89.48	71.05	−1.343	Nuclear

MW: Molecular weight, PI: Protein isoelectric point, II: Instability index, GARVY: Grand average of hydropathicityin.

## Data Availability

The datasets presented in this study can be found in online repositories. The name of the repository and accession number can be found below: “https://www.ncbi.nlm.nih.gov/genbank/ (accessed on 7 July 2022)”, GISP01000000.

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
