# Peer review of "Genome-Wide Identification of bZIP Transcription Factors in Faba Bean Based on Transcriptome Analysis and Investigation of Their Function in Drought Response"

_plants, 2023, doi:10.3390/plants12173041_

Round 1

Reviewer 1 Report

The following corrections need to be addressed before acceptance of the manuscript for publication:

(1) At 4.7, there is no mention of the number of replicates used in the study. The authors need to present detailed information in Materials and Methods

(2) Present the discussion with suitable sub-headings for easy understanding to the readers

Author Response

Dear Editor and Reviewers:
On behalf of the co-authors, I am very grateful to you for giving us the opportunity to revise our manuscript. We really appreciate your valuable comments and suggestions on our manuscript entitled "Genome-Wide Identification of bZIP Transcription Factors in Faba Bean Based on Transcriptome Analysis and Investigation of Their Function in Drought Response" (No. 2578430). We have therefore studied the reviewers’ comments carefully and tried our best to revise our manuscript accordingly. Changes in the revised manuscript are highlighted. Please see below for a point-by-point response to the reviewers’ comments and concerns.

(1) At 4.7, there is no mention of the number of replicates used in the study. The authors need to present detailed information in Materials and Methods

Response: Thank you for your suggestions. We added them in the revised manuscript. See lines 416-418.

(2) Present the discussion with suitable sub-headings for easy understanding to the readers

Response: Thanks to your suggestion, we added them in the revised manuscript.

Reviewer 2 Report

The article is original and will contribute to the science I have the following suggestions to be incorporated in the article.

1. It would be interesting to investigate the expression patterns of the identified VfbZIP genes under other abiotic stresses, such as salinity or heat, to gain a comprehensive understanding of their roles in stress responses.

2. Further studies could focus on characterizing the regulatory mechanisms involved in the activation or suppression of VfbZIP genes under drought conditions, such as identifying specific cis-acting elements and transcription factors that interact with VfbZIP promoters.

3. I would like to suggest articles which have to be cited in the introduction to improved it 1. https://doi.org/10.1016/j.scitotenv.2020.144751 2. doi: 10.1038/s41467-021-22758-0

4. The functional analysis of VfbZIP5 in tobacco provides valuable insights into its potential role in enhancing drought tolerance. It would be worthwhile to explore its effects on other agronomically important crops to assess its broader applicability for improving drought resistance.

5. Investigating the downstream target genes regulated by VfbZIP5 and other identified VfbZIPs could shed light on the specific molecular pathways and processes involved in the drought response in faba bean.

6. Comparative analysis of the identified VfbZIP genes with bZIP genes from other legume crops could provide insights into the evolutionary conservation and divergence of bZIP-mediated stress responses in leguminous plants. The discussion needs to have a comparative discussion. Please see more relevant literature i.e., 1. https://doi.org/10.1016/j.ygeno.2023.110635 2. 

7. It would be valuable to explore the genetic variation and allelic diversity of the identified VfbZIP genes in different faba bean cultivars with contrasting drought tolerance levels to determine their potential as molecular markers for breeding programs.

The English is fine

Author Response

Dear Editor and Reviewers:
On behalf of the co-authors, I am very grateful to you for giving us the opportunity to revise our manuscript. We really appreciate your valuable comments and suggestions on our manuscript entitled "Genome-Wide Identification of bZIP Transcription Factors in Faba Bean Based on Transcriptome Analysis and Investigation of Their Function in Drought Response" (No. 2578430). We have therefore studied the reviewers’ comments carefully and tried our best to revise our manuscript accordingly. Changes in the revised manuscript are highlighted. Please see below for a point-by-point response to the reviewers’ comments and concerns.

1. It would be interesting to investigate the expression patterns of the identified VfbZIP genes under other abiotic stresses, such as salinity or heat, to gain a comprehensive understanding of their roles in stress responses.

Response: Thank you for your valuable suggestion. We just focus on the tolerant response in the manuscript, however, we are planning to investigate the response to other stress in the future as you suggested.

2. Further studies could focus on characterizing the regulatory mechanisms involved in the activation or suppression of VfbZIP genes under drought conditions, such as identifying specific cis-acting elements and transcription factors that interact with VfbZIP promoters.

Response: We appreciate your constructive suggestions. We’ll conduct those work in the future.

3. I would like to suggest articles which have to be cited in the introduction to improved it 1. https://doi.org/10.1016/j.scitotenv.2020.144751 2. doi: 10.1038/s41467-021-22758-0

Response: Thank you for your suggestion. We cited them in the introduction. See references 3 and 33 in the introduction.

4. The functional analysis of VfbZIP5 in tobacco provides valuable insights into its potential role in enhancing drought tolerance. It would be worthwhile to explore its effects on other agronomically important crops to assess its broader applicability for improving drought resistance.

Response: Thank you for your suggestion. We just focus on the drought response at present, so we ignore the observation of other traits. We’ll investigate the drought response and other agronomical traits when VfbZIP5 was overexpressed in important crops in the future.

5. Investigating the downstream target genes regulated by VfbZIP5 and other identified VfbZIPs could shed light on the specific molecular pathways and processes involved in the drought response in faba bean.

Response: Thank you for your suggestion. We are planning to conduct those work in the future.

6. Comparative analysis of the identified VfbZIP genes with bZIP genes from other legume crops could provide insights into the evolutionary conservation and divergence of bZIP-mediated stress responses in leguminous plants. The discussion needs to have a comparative discussion. Please see more relevant literature i.e., 1. https://doi.org/10.1016/j.ygeno.2023.110635 2. 

Response: Thank you for your valuable suggestion. We compared the VfbZIP5 with bZIP genes in other legume crops whose function had been identified as referred to in the suggested paper (see reference 66 in the discussion). The discussion was seen in lines 326-331 and Supplementary Figure 2, Table S6. 

7. It would be valuable to explore the genetic variation and allelic diversity of the identified VfbZIP genes in different faba bean cultivars with contrasting drought tolerance levels to determine their potential as molecular markers for breeding programs.

Response: Thank you for your suggestion. We are planning to conduct those words as your suggestion.

Reviewer 3 Report

Faba bean is an important cool-season edible legume crop that is constantly threatened like any other crop by abiotic stresses. This study, performed genome wide analysis of basic leucine zipper (bZIP) gene family which is one of the most abundant and diverse families of transcription factors in plants. This TFs reported to regulates plant growth and development and plays an important role in the response to biotic and abiotic stresses.

This study comprehensively performed and identified 18 Tfs through genome-wide identification in faba bean based on transcriptome analysis. Also investigated the structural and functional characteristics of these Tfs. Protein-protein interaction (PPI) analysis showed that VfbZIP5 interacted with seven proteins in faba bean

Also developed the transient over expressing tobacco lines and further analysed the functions of one the TFs under by exposing the transgenic lines to drought stress. These overexpressing transgenic lines in tobacco showed improved drought tolerance, with lower levels of proline (PRO), malondialdehyde (MDA), and peroxidase (POD) compared to those overexpressing an empty vector under 10 days of drought stress. This study also further confirmed the expression of the other Tfs and observed higher expressions. These findings clearly shows the role and functions of VfbZIP5 under in drought stress conditions to drought stress and the wider importance of VfbZIPs in plant abiotic stress responses.

However, I have some suggestions to improve the presentation of this manuscript. 

Abstract: Concise and well written

Line 28, enhanced the drought tolerance.

Introduction:

With respect to abiotic stress, tolerance/tolerant word is widely used instead resistance/resistant. Appropriate chages required to be made throughout the text.

Line 50: regulating tolerance mechanism to various abiotic stresses in plants

Introduction part, most of the cited references are related to Arabidopsis, wheat, etc. Some of the references on agriculturally important crops related to this study such as given below needs to be cited

Augustine, S.M., Ashwin Narayan, J., Syamaladevi, D.P., Appunu, C., Chakravarthi, M., Ravichandran, V., Tuteja, N. and Subramonian, N., 2015. Overexpression of EaDREB2 and pyramiding of EaDREB2 with the pea DNA helicase gene (PDH45) enhance drought and salinity tolerance in sugarcane (Saccharum spp. hybrid). Plant cell reports, 34, pp.247-263.

Other crops over expressed with ZIP transcription factor..

Results: Very comprehensive and detailed the results obtained in the study.

Phylogeny tree of VfbZIP in figure 1, 3 & 4 is good way of presenting the result of protein alignment, motif distribution and cis-element distribution, respectively.

Line 194, In Figure 5, what are CK1-16, CK2-16, CK1-64 and CK2-64.

Line 239, OE) and empty vector plants (EV) before and 15 days after drought stress

Line 236-237, Figure 7 (C, D, E & F) Values of Proline, POD, MDA and SOD are higher in empty vector transformed events 10 days after stress. Any possible reasons.

Discussion: Well discussed.

Line 316, ROS, highly reactive radicals produced in response.

Materials and methods:

Line 423, expression of the VfbZIPs as described by Han et al., 2021

Line 430, Include light intensity and relative humidity

Author Response

Dear Editor and Reviewers:
On behalf of the co-authors, I am very grateful to you for giving us the opportunity to revise our manuscript. We really appreciate your valuable comments and suggestions on our manuscript entitled "Genome-Wide Identification of bZIP Transcription Factors in Faba Bean Based on Transcriptome Analysis and Investigation of Their Function in Drought Response" (No. 2578430). We have therefore studied the reviewers’ comments carefully and tried our best to revise our manuscript accordingly. Changes in the revised manuscript are highlighted. Please see below for a point-by-point response to the reviewers’ comments and concerns.

1. Line 28, enhanced the drought tolerance.

Response: Thank you for your suggestion. We corrected it.

2. With respect to abiotic stress, tolerance/tolerant word is widely used instead resistance/resistant. Appropriate chages required to be made throughout the text.

Response: Thank you for your valuable suggestion. We corrected them throughout the text.

3. Line 50: regulating tolerance mechanism to various abiotic stresses in plants

Response: Thank you for your suggestion. We corrected it.

4. Introduction part, most of the cited references are related to Arabidopsis, wheat, etc. Some of the references on agriculturally important crops related to this study such as given below needs to be cited

Augustine, S.M., Ashwin Narayan, J., Syamaladevi, D.P., Appunu, C., Chakravarthi, M., Ravichandran, V., Tuteja, N. and Subramonian, N., 2015. Overexpression of EaDREB2 and pyramiding of EaDREB2 with the pea DNA helicase gene (PDH45) enhance drought and salinity tolerance in sugarcane (Saccharum spp. hybrid). Plant cell reports, 34, pp.247-263.

Other crops over expressed with ZIP transcription factor.

Response: Thank you for your valuable suggestion. We added them. See lines 51-52, 74-79.

5. Line 194, In Figure 5, what are CK1-16, CK2-16, CK1-64 and CK2-64.

Response: Thank you for your suggestion. We added the notes, that is, CK1-16/CK1-64 represents the drought-sensitive variety (E-Can No. 1) after 16 or 64 hours of germination under normal treatment. CK2-16/CK2-64 represents the drought-tolerant variety (CDAS105) after 16 or 64 hours of germination under normal treatment. Changes and additions have been made in lines 200-202.

6. Line 239, OE) and empty vector plants (EV) before and 15 days after drought stress

Response: Thanks for the suggestion, it has been added in line 243.

7. Line 236-237, Figure 7 (C, D, E & F) Values of Proline, POD, MDA and SOD are higher in empty vector transformed events 10 days after stress. Any possible reasons.

Response: The possible reasons why VfbZIP5-OE has lower levels of PRO, POD, and MDA than empty vector transformed events are that drought can result in the enhancement of Proline, POD, MDA, and SOD in both empty vector transformation and VfbZIP5-OE. However, just as the explanation in the discussion section of Lines 324-326, The less increase of MDA and POD levels may be helpful to regulate membrane lipid peroxidation and ROS neutralization to enhance the drought tolerance in VfbZIP5-OE, whereas the PRO level is not tightly related to drought tolerance.

8. Line 316, ROS, highly reactive radicals produced in response.

Response: Thanks for the suggestion, it has been changed in line 323.

9. Line 423, expression of the VfbZIPs as described by Han et al., 2021

Response: Thanks for the suggestion, it has been changed in line 437.

10. Line 430, Include light intensity and relative humidity

Response: Thanks to your suggestion, the light intensity and relative humidity of the greenhouse have been supplemented at lines 444-445.